# Prognostic and Predictive Biomarkers in Familial Breast Cancer

**DOI:** 10.3390/cancers15041346

**Published:** 2023-02-20

**Authors:** Siddhartha Deb, Anannya Chakrabarti, Stephen B. Fox

**Affiliations:** 1Anatpath, Gardenvale, VIC 3185, Australia; 2Monash Health Pathology, Clayton, VIC 3168, Australia; 3Specialist Breast Cancer Surgery, Richmond, VIC 3121, Australia; 4Sir Peter MacCallum Department of Oncology, Peter MacCallum Cancer Centre, University of Mebourne, Melbourne, VIC 3101, Australia

**Keywords:** familial breast cancer, biomarker, *BRCA1*, *BRCA2*, prognostic and predictive

## Abstract

**Simple Summary:**

The importance of identifying and targeting in treating familial breast cancers is becoming increasingly recognized, in particular with increased availability of PARPi-based therapies. This review paper focuses on recognized and emerging prognostic and predictive tumour biomarkers, both recognized and emerging, specifically within familial breast cancer. In particular, these familial breast cancers appear to be different to sporadic lesions, with different interactions and clinical relevance of well-described cancer networks. More so, pre-clinical studies and in vitro models also demonstrate potentially relevant clinical biomarkers, as yet not studied in these groups. Currently, the wider literature has largely focused on prognostic and predictive factors in all breast cancers in general, or biomarkers for genetic susceptibility. This review is novel as there are none currently reviewing biomarkers only within familial breast cancers.

**Abstract:**

Large numbers of breast cancers arise within a familial context, either with known inherited germline mutations largely within DNA repair genes, or with a strong family history of breast and/or ovarian cancer, with unknown genetic underlying mechanisms. These cancers appear to be different to sporadic cases, with earlier age of onset, increased multifocality and with association with specific breast cancer histological and phenotypic subtypes. Furthermore, tumours showing homologous recombination deficiency, due to loss of *BRCA1*, *BRCA2*, *PALB2* and *CHEK2* function, have been shown to be especially sensitive to platinum-based chemotherapeutics and PARP inhibition. While there is extensive research and data accrued on risk stratification and genetic predisposition, there are few data pertaining to relevant prognostic and predictive biomarkers within this breast cancer subgroup. The following is a review of such biomarkers in male and female familial breast cancer, although the data for the former are particularly sparse.

## 1. Introduction

Breast cancer is the most common cancer seen, affecting 1 in 8 women worldwide, and also contributing to 15% of all cancer-related deaths [1]. Within this population, approximately 5–10% are described as familial breast cancers, with a strong family history of breast or ovarian cancer, and frequently carrying pathogenic germline mutations [2,3]. Of these, the most commonly seen are in the Breast Cancer gene 1 (*BRCA1)*, Breast Cancer gene 2 (*BRCA2)*, Checkpoint kinase 2 (*CHEK2)* and partner and localizer of the BRCA2 gene *(PALB2)* tumour suppressor genes. These account for 50% of cases of familial breast cancer and are the most-characterised clinical and biological cohorts to date.

It is increasingly recognised that familial breast cancers show multiple differences compared with sporadic breast cancer in both men and women, and form the clinical and biological standpoints [4]. These tumours often show strong geno-phenotypic correlation, with *BRCA1* carriers showing increased representation of basal phenotype cancers in women [4], and *BRCA2* carriers showing increased representation of micropapillary-type cancers in men [5]. Familial breast cancers show more frequent early onset and multifocality, with an increased incidence of bilateral cancers. Specifically, in familial breast cancers, inactivation of the DNA damage repair pathways has been shown to alter prognosis, which has been exploited in some newer treatment modalities focused on targeting sensitivity to DNA-damaging therapies and immunotherapies.

Women who harbour these mutations have a 45–84% lifetime risk of breast cancer [6,7]. There is some clinical and pathological heterogeneity, with variation seen in the age of onset, hormonal status and cancer type, as well as treatment outcomes. While the majority of research has focused on breast cancer susceptibility and risk modification, there is a paucity of data on potential biomarkers and predictive factors in these cancers, and whether further tailoring of therapy may be needed in these patients. The following is a review of studies and potential prognostic and predictive biomarkers in familial breast cancer.

## 2. Prognostic and Predictive Biomarkers

### 2.1. BRCA-Associated Homologous Recombination

The use of Poly ADP Ribose Polymerase (PARP) inhibitors (PARPi) has radically improved the treatment outcomes of *BRCA1*/2mutation (mut) breast cancer patients. The inhibition of PARP leads to an inability to recruit the base excision pair (BER) machinery for single-strand DNA break repair, with the consequent result of a DNA double-strand break formation (Figure 1) [8]. In cells with proficient Homologous Recombination (HR), this is subsequently repaired with normal function maintained. In tumours with *BRCA1*/2mut and no HR function, the cells undergoes cell death or are forced to rely on highly inaccurate classical non-homologous end joining (cNHEJ), resulting in eventual genomic instability and cell death (Figure 1) [9]. Clinical trials of PARPi showed anti-cancer efficacy with the approval for Olaparib and talazoparib monotherapies for locally advanced/metastatic deleterious/suspected deleterious germline *BRCA*mut, Human Epidermal growth factor Receptor (HER2)-negative breast cancers [10,11].

Several mechanisms of PARPi have now been described that account for the treatment resistance and tumour growth observed both in vitro and in vivo. Similar to other therapies, decreasing availability of intracellular PARPi may occur through up-regulation of drug efflux transporter genes, such as ATP Binding Cassette Subfamily (*ABC*)*-A1a/b* and *ABCG2* [12]. While shown in mouse models [13], as yet, there are no studies describing *ABCB1* levels in PARPi-treated *BRCA1*/*2mut* patients and, therefore, it is unclear whether this may be a potential biomarker of treatment resistance, and whether coadministration of Multi-Drug Resistance 1(MDR1) inhibitors with PARPi treatment might be a possible future strategy in these patients.

Alterations to the binding of PARPi have also been observed, with numerous mutations in PARP1 resulting in a loss or decreased trapping ability of PARPi [14]. In some instances of *BRCA1*/2mut, where there is still some functional HR, *PARP1* mutations, which reduce PARP1 binding affinity to DNA or disrupt PARylation via disruption of PAR removal, may also lead to acquired PARP inhibitor resistance [14,15].

Cancer cells may also abrogate PARP loss by restoring HR repair or by re-establishing replication fork stability [16,17,18,19]. HR restitution may occur with reversion mutations in HR genes, resulting in functioning HR proteins, which has been observed in tumour samples from patients with acquired resistance to PARPi. This can even occur with expression of hypomorphic mutant proteins that are still capable of RAD51-mediated DNA end resection, formation of D-loop and subsequent repair [20]. The reliance of cyclin-dependent kinases (CDKs) on DNA end resection also restores HR, through ATR activation and phosphorylation of CtIP [21] and the Mre11/RAD50/Nbs1(MRN) complex [22]. In vitro studies have showed reversal of PARPi resistance in *BRCA*mut models, with a case study showing improved efficacy of PARPi combined with a CDK4/6 inhibitor (palbociclib) compared with PARPi alone in treating *BRCA*mut, ER-positive breast cancers [10]. In addition, defects in CDK accessory factors, such as 53BP1, Shieldin complex and rvesionless 7-like (REV7), activated by Ataxia-Telangiectasia (ATM), appear to mediate p53-binding protein 1 (53BP1)-dependent DNA repair in PAPRi resistance [23,24]. Limited studies have been performed on these biomarkers in breast cancers, with uncertain prognostic potential, and most likely suggestive of different functionality between tumours that are HR-proficient versus -deficient tumours [25,26].

In a recent study of 308 *BRCA1*/2mut tumours, interrogation of non-*BRCA* DNA Homologous Recombination Repair (HRR) genes showed no obvious effect on talazoparib efficacy of *BRCA* Loss of Heterozygosity (LOH), effect of HRR gene mutational burden and tumour homologous recombination deficiency, as assessed by genomic LOH (gLOH)] [27]. Several study limitations were noted, however, and although there was high concordance between germline and tumour *BRCA1*/2mut, with 82.6% LOH of *BRCA1*/2, there was unknown methylation status of *BRCA1*/2 wild-type (wt) and HRR genes, and a lack of power due to small patient numbers in some subgroups.

DNA repair due to increased loading of RAD51 appears to be a critical step in the HR pathway. Indeed, using RAD51 as a biomarker for PARPi has been shown to be biologically relevant, with an immunofluorescence-based RAD51 assay showing capability to identify HR proficiency in both pre-clinical and clinical samples [28,29]. Practical limitations of some assays, however, were present, as the methodology utilized irradiated live tumour cells [30], which are often not available in clinical specimens. Nevertheless, transfer of the RAD51 assay on formalin-fixed samples without fresh tissue have shown some promise, with an ability to identify HR-deficient tumour cells that were sensitive to PARPi therapy [28,29,31].

With the promise of increased tumour efficacy of PARPi in combination with immunotherapies and other molecular targets, including the PI3K/AKT/mTOR pathway and other HR mechanisms seen in pre-clinical models, there are likely to be further biomarkers identified to aid more specific therapies.

### 2.2. Non-BRCA-Associated Homologous Recombination

As mentioned, there is a paucity of studies evaluating biomarkers, specifically in familial breast cancers, even in *BRCA*-associated tumours. For non-*BRCA*-associated tumours, the data are even more sparse. A single study examined RAD21, a key protein in the cohesin complex required for chromosome segregation and high-fidelity DNA repair by homologous recombination. The study of 94 familial breast cancers [32] demonstrated RAD21 overexpression in 43%, 44% and 33% of *BRCA1*, *BRCA2* and *BRCA*X tumours, respectively, and expression was associated with shortened cancer-specific overall survival (Hazards Ratio 1.66, *p* = 0.003). In particular, when stratified for RAD21 expression, there were significantly shorter survival outcomes in chemotherapy-treated patients (*p* = 0.036) but not those treated with hormone therapy (*p* = 0.881). Expression was prognostic in *BRCA2mut* (*p* = 0.006) and *BRCA*X (*p* = 0.008) subsets but not in *BRCA1*mut cancers (0.71). This suggests a mechanism of chemotherapy resistance in *BRCA2* and *BRCA*X cancers, which may be abrogated in *BRCA1*mut tumours, most likely due to inherent inactivation of RAD21, as *BRCA1*wt has been shown to be required to activate RAD21 function [33]. 

Similarly, a study was performed on HuR, an mRNA binding Hu family protein that enhances the regulation translation of target transcripts through enhancing their stability [34]. In many cancers, including pancreatic exocrine adenocarcinoma, ovarian carcinoma, non-small cell lung carcinoma, colon carcinoma and upper genito-urinary carcinomas [35], HuR levels are increased and oncogenic, primarily activated by mitogen-activated protein kinases (MAP-Ks). In a study of 623 familial breast cancers [36], HuR protein expression was found to be prognostic in *BRCA*X patients (*n* = 525). Cytoplasmic HuR expression was higher in *BRCA*X patients (39%) when compared with sporadic tumours (39%) and was associated with ER and PR negativity, p53 immunopositivity, high tumour grade and ductal histology. Notably, HuR levels were even higher in *BRCA1* (63%) and *BRCA2* (62%) tumours and associated with a non-significant reduction in survival in *BRCA2* mutation carriers. There was no prognostic effect in the *BRCA1* group. Based on these findings, HuR appears to be involved in the oncogenic pathways of familial breast cancer, probably through activation of the MAP kinase pathway, and it is a prognostic factor in at least some subsets of familial breast cancer.

While not specifically looking at familial cancers, a study by Takada et al. [37] examined breast tumours with *BRCA1* or *BRCA2* alterations (methylation and/or loss or uniparental disomy) and showed significant associations with higher numbers of genomic aberrations and higher percentage of TP53 alterations than *BRCA1*/2wt tumours. Although there was no difference in overall survival between patients with or without *BRCA1*/2 alterations, when stratifying by *PALB2*, *PAGR1*, *RAD51B*, *FANCM*, *MLL4* or *ERCC1*/2a mutations, a small number of *BRCA1*- patients (*n =* 27) [37], with an additional defect in HR genes, had a significantly shorter survival. The combination of these HR defects was hypothesised to result in chemoresistance and, thus, the poorer outcomes in patients with *BRCA1*-altered breast cancer, and may show a similar correlation in familial breast cancers.

### 2.3. HER2

Within breast cancer, the identification of HER2 amplification as both a marker for intrinsic phenotypes as well as a target for specific therapies has stratified treatment protocols as well as patient outcomes [38,39]. As part of the ERBB/HER/EGFR family, amplification of this oncogene is seen in ~15% of all female breast cancers and ~5–9% of male breast cancers. HER2 amplification is associated with a more aggressive phenotype and shorter disease-free survival (DFS) and overall survival (OS) [39]. Multiple lines of therapy, including monoclonal antibodies, tyrosine kinase receptor inhibitors (TKI), signal transduction inhibitors and antibody drug conjugates, have been demonstrated in multiple trials, including at all stages, to be effective in breast cancer, albeit with inferior outcomes overall to HER2-negative tumours [40].

The characterisation of an HER2-positive tumour in familial breast cancer is still poorly described, with few large studies performed. Nevertheless, there is a reduced incidence of HER2-positive breast cancers in *BRCA1-* and *BRCA2*-associated tumours, with a recent pooled analysis of a total of 21,083 *BRCA1* and *BRCA2*mut cancers from 73 studies [41], showing mean HER2-positive rates of 8.3% and 10.3%, respectively (combined 9.1%, 95% CI interval 7.3–11.2%). Notably, even fewer studies have characterised HER2, variably determined by either FISH or immunohistochemistry within non-*BRCA1*/2 (so-called *BRCA*X) familial breast cancers, with nine small studies showing a wide range of HER2 positivity (9–60%) [42].

The significance of the HER2 oncogenic drive in familial breast cancer is most likely heterogenous. While there are yet no studies determining the oncogenic drive of simultaneous mutations in *BRCA1*/2 and HER2, there are likely tumours that either show combined *BRCA1*/2 and HER2 mutation, or HER2 amplification with an intact *BRCA1*/2 pathway. Perhaps evidence of this is seen in some larger studies [43] that have shown that HER2-positive tumours within the *BRCA* population are more frequently associated with oestrogen positivity (75% vs. 46%, *p* < 0.0026), with a trend towards an older age of onset (mean 48 years vs. 42 years). Notably, there is also a tendency in *BRCA1*/2 mutation carriers who have had multiple tumours to show higher rates of HER2 positivity in later-onset lesions (78%), with a case report in a *BRCA1*mut carrier showing loss of the *BRCA1*mut gene, without inactivation of the *BRCA1*wt gene presenting with an HER2-amplified cancer [44].

A recent case-matched analysis of 700 breast cancers [45], stratified by HER2 tumour positivity and *BRCA* mutation status, showed significant clinical and pathological difference between HER2-positive *BRCA1*/2mut and HER2-negative *BRCA1*/2mut tumours, with a higher proliferative index (Ki67) and grade in the HER2-positive cancers. There was no difference in the HER-positive *BRCA*wt and HER2-positive *BRCA*mut groups. The HER2-positive *BRCA*mut subgroup showed overall worse survival compared with HER2-negative and *BRCA*wt tumours (Hazards Ratio = 3.4, 95% 1.3–16.7) but appeared not to show a significant difference to the HER2-positive *BRCA*wt or HER2-negative *BRCA*mut subgroups.

As of yet, while there is a theoretical advantage in combined pharmacological therapy in HER2-positive tumours occurring in *BRCA1*/2 mutation carriers, there is no definitive guidelines or data pertaining to this population. In preclinical models, a response to HER2, TKIs were seen in tumours with inactivated *BRCA2* mutations. Furthermore, the addition of PARPi showed an enhanced effect and showed an advantage to combination therapy in this instance [46], suggestive of an advantage of dual therapy.

### 2.4. Hypoxia

Hypoxia-Inducible Factors (HIFs) and their regulatory molecules regulate multiple pathways through the transcription of genes that allow for cellular adaption to changes in oxygen tension, especially hypoxic environments [47]. Examples when HIF-1α is overexpressed include established downstream targets, such as carbonic anhydrase IX (CAIX) and glucose transporter-1 (Glut-1) [48]. HIFs are involved in the pathogenesis and progression of many cancer types, including triple-negative breast and BRCA1-associated breast cancer, as they provide a growth advantage to rapidly growing tumours where angiogenesis may not be adequate. HIF-1α overexpression correlates with poor prognosis in triple-negative breast cancer (TNBC), including familial-associated breast cancer [49,50,51,52,53,54,55].

There are some data to suggest that *BRCA1* modulates the hypoxic response by regulating HIF-1α stability and, thereby, vascular endothelial growth factor levels (see below) [55]. Furthermore, possibly due to limiting angiogenesis, hypoxia-induced HIF-1α overexpression has been reported to be more than 2-fold more frequent in *BRCA1*-mutation-associated breast cancer than in sporadic breast cancers [56]. Conversely, *BRCA2*-mutation-related breast cancers express HIF-1α less frequently [55]. It has been proposed that the HIF pathway is integral to the progression of *BRCA1*-associated breast cancer and may be a target for emerging targeted therapies directed to this pathway and downstream targets [57].

### 2.5. Vascular Endothelial Growth Factor (VEGF)

As above, VEGF is an important HIF-upregulated gene, which plays a pivotal role in angiogenesis. VEGF has been shown to be important in breast carcinogenesis, with increased expression of VEGF in HER2, luminal B and basal-type breast cancers [58], and being an adverse prognostic factor for some breast cancer subsets [58]. In a study of 60 breast cancers [59], stratified by familial history and known *BRCA* status, there were significantly higher levels of VEGF and Angiopoietin-1 and Angiopoietin-2, two other angiogenic factors involved in vessel remodelling in patients with *BRCA1*/2 mutations. This suggests that VEGF with Tie2 signalling could stimulate sprouting angiogenesis in *BRCA*-related cancer, accounting for the high microvessel density often observed in these cancers (especially the triple-negative/basal-type breast cancer common in *BRCA1* carriers) and suggest these breast tumours might be attractive targets for anti-angiogenic therapeutics.

### 2.6. Cell Cycle Regulation

Through differential gene expression analysis, Severson et al. [60] showed that the most common genes that were upregulated in *BRCA1*-like tumours were centred on the FOXM1 network. FOXM1 is a member of the forkhead superfamily transcription factors and a key regulator of cell cycle progression and DNA damage repair. FOXM1 is over-expressed in most human cancers, including epithelial cancers, such as prostate adenocarcinomas, gastrointestinal cancers, non-small-cell lung cancers and high-grade ovarian cancers, and is predictive of poorer survival in breast cancer. While only a small proportion of these *BRCA1*-like tumour harboured *BRCA1* germline mutation, the finding that FOXM1 and downstream factors CDK4/6 are highly expressed in *BRCA1*-like tumours may have predictive implications in the setting of available CDK4/6-based therapies in *BRCA1*/2mut cancers.

### 2.7. Androgen Receptor

The androgen receptor (AR) is a steroid signalling family member and functions as a transcription factor. With ongoing development of AR-targeted therapies, including bicalutamide and ezalutamide (both nonsteroidal antiandrogen), showing clinical benefit [61,62], AR is a potentially useful biomarker for these therapies. It is more commonly associated with hormone-receptor-positive breast cancer, where AR expression is seen in >70% of tumours, and some small studies have shown AR expression in 30% (*n =* 13 of 43) of *BRCA1*mut and 78% (*n =* 14 of 18) of *BRCA2*mut tumours [63]. However, there is a complexity to the significance of AR expression in breast cancer, with studies showing that the prognostic and predictive power is dependent on the molecular subtype of the tumour, and other factors such as EGFR [62]. In TNBC subsets, AR positivity is associated with improved features, such as lower tumour grade, risk of nodal involvement and lower proliferative index. While AR is associated with improved DFS in TNBC, interestingly, AR expression may imply less sensitivity to chemotherapy [64]. Nonetheless, as ~20% of ER/PR-negative *BRCA1*mut breast cancers express AR, the modulation of AR may offer new treatment options for these high-risk cancers. Further study into the mechanism of AR function in familial breast cancers is needed, with some studies into prostate cancers showing improved efficacy of PARPi in conjunction with antiandrogen therapies, albeit in HR-proficient cell lines [65]. There are, as yet, no clinical trials on antiandrogen therapy in familial breast cancer.

### 2.8. Epidermal Growth Factor Receptor

While the high frequency of the epidermal growth factor receptor (EGFR) is well recognised in triple-negative breast cancers [66], and has been shown to be an independent prognostic factor disease and overall survival [67,68], paradoxically, in one report, low EGFR expression was observed in *BRCA*mut tumours compared with other TNBCs [69]. Notably, there are data to suggest the blockade of EGFR may alter DNA damage repair mechanisms, with decreased induction of Rad51 in DNA repair, and shuttling of BRCA1 protein to the cytoplasm in the cell in vitro [70]. Promisingly, in pre-clinical TNBC models, a combination of EGFR inhibition with PARPi-targeted therapies also showed synthetic lethality through abrogation of the HR-dependent mechanism [71]. Nevertheless, we were unable to identify, at the time of writing, studies that have evaluated EGFR as a prognostic marker in familial breast tumours, or the benefits of EGFR-targeted tyrosine kinase inhibitors, specifically in clinical familial breast cancer studies.

### 2.9. MicroRNA (miR)

MiR binds mRNA and, through this post-transcriptional mechanism, regulates DNA expression. They are aberrantly expressed in tumours [72] and, being stable, are able to be extracted in tissue, but also isolated from fluids, such as blood [73], saliva [74] and urine [75].

Some miRNAs have been used to stratify risk in patients with breast/ovarian cancer and are associated with *BRCA1/2* mutations [73,76]. Different miRNA expression profiles have been identified in healthy women, women with sporadic breast cancer and women with *BRCA*-mut breast cancer [75,77], and, to some degree, between *BRCA1-*, *BRCA2-* and *BRCA*X-related tumours. Although only a small number of specimens was studied, Murria-Estal et al. [78] reported 15 differentially expressed miRNAs (miR 4756-5p, miR 1273c, miR 4519, miR 323-3p, miR 4731-5p, miR 4498, miR 4417, miR 4783-3p, miR 129-5p, miR 4680-3p, miR 583, miR 206, miR 423-3p, miR 1181, miR 3169) that differentiate between *BRCA1*, *BRCA2*, *BRCA*X and sporadic breast tumours, albeit with relatively low accuracy (75%). Yan et al. showed an miR signature correlating with reduced/negative staining for downstream protein FOXP1, Cyclin D1 and NRP1 able to predict germline *BRCA1* mutation status with a sensitivity of 92%, specificity of 44%, positive predictive value of 38% and a negative predictive value of 94% [79]. Other classifiers based on variable numbers of miRNAs have been suggested to distinguish *BRCA1*/2-mut (*n =* 6) and hereditary breast cancers (*n* = 15) from non-mutated breast tumours, respectively, with relatively high accuracy [80,81]. However, validation studies of these are lacking.

Notably, over 100 miRNAs have been shown to be interactive for the *BRCA* proteins. While prognostic significance has not been seen specifically in familial breast cancer cohorts, studies in TNBC have identified several miRNAs that may show potential significance in *BRCA1*/2mut cancers. These include elevated levels of miR-21, miR-27a/b, miR-210 and miR-454, associated with shorter OS and high miR-548c-5p and high miR-29b-1-5p, associated with improved overall survival in breast cancer patients with basal or TNBCs [82,83,84,85,86]. Furthermore, low expression of miR-155 [87] was associated with poor overall survival in TNBC patients, and low expression of miR-374a/b [88] and high miR-214 and miR-454 expression correlated with shorter disease-free survival [83,89]. These miRNAs appear to function across a wide range of cellular mechanisms that include cell proliferation, apoptosis and immunoregulatory mechanisms. Direct associations are seen between the has-miR-548 family, which has binding sites in *BRCA2*, and epigenetic control of miR-155 by *BRCA1*.

With regard to predictive makers, several miRNAs, including miR-146a, miR-146b-5p and miR-182, have been reported to reduce BRCA1 protein expression, with miR-182 expression also showing sensitivity to PARPi in cancer cell lines [90,91]. Other studies have reported that miR-493-5p overexpression is able to restore genomic stability in *BRCA2*-mut cells, leading to acquired resistance to PARP inhibitors [92]. In vivo studies also suggest miR-664b-5p may be important in increasing chemosensitivity in *BRCA1*mut TNBC [93]. MiR-664b-5p, by targeting CCNE2 (a G1-cyclin binding CDK2) protein expression, acts as a tumour suppressor and increases sensitivity to PARP inhibitors. Indeed, MiR-664b-5p inhibited tumour growth compared with the control in tumour xenograft models, and CCNE2 expression was also inversely correlated with miR-664b-5p expression in 90 TNBC patient samples [93].

Recent studies showed that results of the treatment with PARP1 inhibitors in breast and ovarian cancers may be dependent on high expression, particularly for miRNAs, and low expression of BRCA1 [94]. In mouse models, increased expression of miRNA-9 inhibited tumour growth during treatment with a PARPi [94]. Using prediction algorithms, Moskwa et al. found miRNA-182 targeted *BRCA1* in breast cancer [91], and overexpression of miRNA-182Its in MDA-MB231 cells was significantly more sensitive to PARPi [91].

In a setting of marked clinical and tumour heterogeneity, the use of miRs as both biomarkers and in therapy is highly appealing. These targets may be especially useful serum interval markers of treatment response and as early markers of progression to advanced disease. While there are no miR-based therapeutics available currently, there may be potential clinical application in various subsets of familial breast cancer. As a result of further investigation in this area, there is also potential for identification of relevant downstream gene and protein biomarkers in familial breast cancers, and increased use of utility of these biomarkers in directing therapies, with several clinical discovery studies underway and one validation clinical trial looking at the predictive value of treatment response to HER2 therapies also present (NCT02656589) [95]. As of yet, there are no miRNA-based clinical studies specifically focused on familial breast cancer.

### 2.10. Single-Nucleotide Polymorphisms (SNPs)/Single-Nucleotide Variants (SNVs)

Several studies have examined the potential risk modification of SNP/SNVs in familial breast cancers and, in particular, *BRCA1*, BRAC2, *CHEK2*, *PALB2* and *ATM* mutation carriers. They have shown both a correlative and inverse association between SNP/SNV panels and risk of breast cancer when compared to the general population, suggestive of the importance of genetic context in potential risk stratification. There are, however, no SNV-based predictors for tumour outcome in familial cancer patients [96].

### 2.11. Commercial Expression Profile Assays

Arguably among the best studied and strongest predictive and prognostic tools in breast cancer, several small studies have reported on the application of molecular assays, such as Oncotype DX^®^, MammaPrint^®^, Prosigna^TM^ and EndoPredict^®^, in hereditary breast cancers [97].

The use of a Breast Recurrence Score (RS) in the Oncotype DX^®^ assay has been shown to be prognostically significant and is a useful predictive too in the treatment of early-stage hormone-receptor-positive breast cancer. Not surprisingly, compared with the general population, both *BRCA1-* and *BRCA2*-associated tumours showed higher mean RS scores (oncotype DX^®^ 23.5-29 vs. 16) in 32 and 33 patients (25 *BRCA2* and 8 *BRCA1*) [98,99] or a higher proportion of intermediate (18–30) or high (>30) RS scores (*BRCA1*—87.6%, *BRCA2*—82.8% vs. general population—46.6%, *p* < 0.001). The RS scores showed an association with higher tumour grade, and more frequent PR-negative status, but did not appear to be influenced by nodal status [100,101].

From a prognostic standpoint, there was no evidence of a statistically different OS between *BRCA* mutation carriers and the general population when stratified by RS score. Notably, while there were differences in treatment between the *BRCA* mutation carriers and the whole general population (with higher levels of chemotherapy in *BRCA*mut patients (54.5% vs. 32.6%, *p* = 0.0090) and lower rates of hormone therapy (81.8% vs. 91.7%, *p* = 0.489) and radiation therapy (24.2% vs. 48.5%, *p* = 0.0065)) [100,101], when matched by RS scores, the treatment protocols appear similar, with no significant differences.

As *BRCA*mut tumours are more frequently treated with chemotherapy and mastectomy, the utility of these tests may be somewhat arbitrary in treatment decision making. However, there is still an uncertainty as to whether these tests may provide similar prognostic and predictive value similar to that seen in the general population [99], or identify potential hormone-positive *BRCA*mut tumours, most likely of low RS that may not need chemotherapy, a group which may comprise an estimated 8–44% of hormone-positive *BRCA*mut tumours.

Currently, there are no studies on the utility of MammaPrint^®^, Prosigna^TM^ and EndoPredict^®^ in familial breast cancer. As the homologous recombination machinery is not evaluated by these assays, newer assays for homologous recombination deficiency may be of predictive use. Several models have been studied to determine the *BRCA* level of tumours and, in particular, those that may benefit most from platinum-based chemotherapy. Vollebergh et al. used an aCGH classifier to identify *BRCA1*-like tumours [102]. The majority of these tumours showed either *BRCA1* mutation or methylation (63%) and improved benefit from platinum-based chemotherapy compared to conventional chemotherapy (Hazard Ratio—0.12, *p* = 0.006).

Several newer platforms have evaluated molecular signatures specific to HR deficiency, and the ‘BRCA likness’ of cancers. At this point, there are up to 49 different mutational signatures described, with many showing an association with specific mutations and pathways [103,104]. The HR-deficiency-specific pattern, “Signature 3” is seen with *BRCA1/2* loss in several cancer streams, including breast, ovarian, prostate and gastric [103]. This signature may be identified by whole-genome sequencing and contemporary bioinformatics tools, such as SigMA [105], able to detect Signature 3 profiles from sequence data procured form large targeted panels used in clinical practice and bypassing whole-genome analysis.

Several commercial assays are now available for HR deficiency, of which two are approved by the FDA for ovarian cancers. Using formalin-fixed paraffin-embedded (FFPE) tissue, the Foundation Medicine’s FoundationFocus CDxBRCA LOH [106] and Myriad’s myChoice^®^ CDx Plus assay [107,108] have been approved as companion diagnostics for identifying patients for rucaparib and olaparib treatment, respectively. The assays identify focused HR gene mutations and variable genomic changes, primarily through LOH for Foundation Medicine and, additionally, large-scale transitions and telomeric allelic imbalance for the Myriad assay, undertaken by targeted panel testing.

Other commercial panels include the SOPHiA Genetics^TM^ DDM Homolgous Recombination Deficiency Panel using targeted next-generation sequencing combined with low-pass whole-genome sequencing to identify mutations in 28 HR-pathway-associated genes and large-scale copy number changes, indicative of HR deficiency. The assay shows high concordance with Myriad myChoice^®^ [109] and can be performed on FFPE or fresh-frozen material from breast cancers. The AmoyDx^®^ HRD Focus Panel can be performed on FFPE tissue and evaluates for pathogenic mutations in *BRCA* genes, as well as genomic changes through SNPs, distributed evenly within the genome. HRDetect, developed by Zainal et al., is a weighted model combing six distinguishing mutational signatures using an assay that was able to accurately identify a *BRCA1*/2-deficient tumour with 98.7% accuracy [110]. This identified all tumours with germline *BRCA1*/2 loss (22 cases out of 560), 22 further cases of somatic *BRCA1*/2 loss and 47 tumours with *BRCA1*/2 deficiency where no mutation was detected, but it was limited by the requirement of whole-genome sequencing for data input.

### 2.12. PIK3CA

Of the limited descriptive studies to date in familial breast cancer, when compared with sporadic tumours, there is a markedly reduced frequency of somatic *PIK3CA* mutations in *BRCA1*/2mut tumours (5–16%) [111,112]. The clinical significance of this is unknown but may be of interest, as *PIK3CA* inhibition has been shown to result in HR deficiency. In particular, *BRCA*-proficient TNBCs have shown increased sensitivity to PARP inhibition following *PIK3CA* blockade. Furthermore, in ovarian cancer cells lines, dual blockade of PI3K and PARP in vitro resulted in downregulation of PI3K/AKT/mTOR signalling and impaired DNA damage response with HRR deficiency, with associated reductions in BRCA, in a setting of intact *PIK3CA*wt and *BRCA*wt [113,114,115].

### 2.13. Immunotherapy Biomarkers

Almost all known moderate and high-penetrance familial breast cancer tumour suppressor genes are associated with the maintenance of genomic stability and its integrity [116,117]. While dysregulation of these cancer pathways is integral to tumour development, as a by-product of gene mutation, chromosomal rearrangement and genomic instability, there appears to be increased tumour immunogenicity through multiple mechanisms, albeit with immune regulatory mechanisms also at play [116,117].

### 2.14. Tumour-Infiltrating Lymphocytes

The role of tumour-infiltrating lymphocytes (TILs) in breast cancer has been studied over the last decade. Notably, high numbers of TILs are frequently present in triple-negative breast tumours and also in HER2-positive cancers. Similar to the association seen in microsatellite unstable cancers, such as colorectal or endometrial carcinomas, loss of homologous recombination repair pathways, frequently seen in familial breast cancers, is thought to increase tumour neoantigens, with cancer models, particularly *BRCA1* deficiency, showing increased somatic mutation loads [118]. Nevertheless, HRR deficiency is not always associated with high tumour mutational burden [119].

Overall, high stromal TILs appear to be predictive for higher rates of pathological complete response (pCR) in the setting of neoadjuvant chemotherapy and prolonged overall survival in certain subsets of breast tumours. Several large studies have shown increased rates of TIL-high tumours in *BRCA1/2* carriers when compared with all WT *BRCA* breast cancer patients [120]. Notably, however, there appears to be a phenotypic bias, with the difference not always observed between *BRCA*mut and *BRCA*wt tumours, at least in triple-negative tumours (*p* = 0.391, *p* = 0.36) [121]. 

There are a few limited studies reporting on the prognostic and predictive relevance of TILs as a biomarker in only familial breast cancers. A study by Sonderstrup of 411 *BRCA1*/2mut early breast cancers showed that stromal TILs increasing in 10% intervals were significantly associated with OS (Hazard Ratio 0.92, 95% CI 0.84–1.00, *p* = 0.05) in *BRCA1* and *BRCA2*mut breast cancers [122]. The association in the *BRCA1* subset was a 10% reduction and a 13% reduction in risk of DFS events with each 10% increment in stromal TILs, even after adjustment for ER status. However, no significant association with survival was observed in the *BRCA2* subgroup by itself. Using almost the same cohort, in 414 *BRCA1*/2mut cancers, Jorgensen et al. [123] examined other immune markers and observed a 26% reduction in risk of disease-free survival for a 10% increase in CD8-positive cells, with a similar trend seen for CD4 and FOXP3 expression, the latter most prominently in *BRCA1*mut-associated tumours. Mortality rates showed a 28%, 46% and 12% reduction for each 10% increase in CD4 and CD8 expression and for each 1% increase in FOXP3 expression, respectively. Interestingly, many of these associations are also seen in studies of TNBCs [124,125,126]. Nevertheless, some studies have observed a direct correlation between increased numbers of FOXP3 T-regulatory cells and improved mortality, which is somewhat unexpected given the immune regulatory role of these cells, and is contrary to the association with poor survival in breast cancer studies unselected for familial breast cancers [127].

Examination of specific T-cell subsets, including T cells (CD3, CD4, CD8), B cells (CD20) and checkpoints (PD-1, PD-L1, see below), to date, has shown no difference between *BRCA1*mut and *BRCA2*mut tumours, and between *BRCA*mut and *BRCA*wt tumours, albeit with the latter studies almost universally performed in triple-negative breast cancers [120,128]. Interestingly, this is somewhat different to what in seen in prostate cancer, where *BRCA2*mut tumours showed increased ratios of intratumoral to extratumoral immune cells, and lower CD8:FOXP3 ratios when compared with *BRCA1*mut and *ATM*mut cancers, suggestive of at least the possibility of a more suppressed microenvironment in some tumour types when stratified by *BRCA* mutation status [129].

A recent study by Grandal et al. is the only to examine TILs in the setting of pre- and post-neoadjuvant therapy in a large patient cohort with *BRCA* germline status available [121]. Similar treatment responses were seen in triple-negative and HER2 tumours when stratified for *BRCA*mut versus *BRCA*wt. There was a markedly superior pCR rate in the luminal *BRCA*mut tumours when compared to *BRCA*wt lesions (33% vs. 5%, *p* = 0.006). Most notably, in this group, there was also a high percentage of post-neoadjuvant stromal TILS in the *BRCA*mut luminal cancers when compared with wild type (*p* = 0.0091). Such a difference was not observed in the TNBC and HER2 subgroups. Notably, there was no difference between any of the three phenotypes (luminal, TNBC and HER2) between the *BRCA*mut and *BRCA*wt tumours in the pre-neoadjuvant biopsy.

### 2.15. Programmed Cell Death Ligand-1 (PDL-1)

Within cancer immunology, there are now well-described immunomodulatory and suppressive mechanisms that allow potentially highly immunogenic tumours to survive in anergic micro-environments. Key regulators within this area are PD-1 (Programmed Cell Death Protein 1) (CD274 or B7-H1) and PDL-1. The transmembrane glycoprotein PD-1 is strongly expressed by activated T cells and in tumour-specific T cells, B cells and NK. [130]. 

When the receptor activates the ligand, regulatory and anergic mechanisms are initiated, most importantly resulting in inhibition of normal activation and proliferation of tumour-specific T cells, thereby abrogating the PI3K/AKT/mTOR and Ras pathways and consequently the cell-killing effect of tumour-specific T cells [131,132]. Downstream inhibition of pro-inflammatory factors and reduced antigen presenting of dendritic cells also occur [133].

In breast cancer, PDL-1 and PD-1 have been best studied in triple-negative breast cancers. Expression appears to be higher both within tumour and immune cells of triple-negative tumours when compared with hormone-positive lesions [134]. In some studies, an association between tumour PDL-1 expression and prognostic factors, such as higher grade, larger tumour size and younger age of onset, have been reported; however, reproducibility and guidelines for robust biomarkers have been hampered by the use of different PD-L1 antibodies raised against different epitopes, different immunohistochemical staining platforms and detection methods and a mix-and-match approach to score immune cells versus tumour staining (Figure 2), and with probable tropism bias, with liver and nodal lesions consistently showing higher levels of PDL-1 tumour expression compared to other sites [135,136]. 

To date, PD-L1 inhibition has shown clinical benefit in triple-negative breast cancer in some studies. IMpassion130 [137], the first randomized controlled study in first-line therapy in unresected metastatic TNBC, showed a significant prolonged progression-free survival in patients receiving combination nab-paclitaxel and atezolizumab (anti PD-L1) compared with nab-paclitaxel alone, the strongest effect in patients in tumours with >1% staining of immune cells (ICs)(PD-L1 positive). However, IMpassion131 [138], which had a slightly different trial design, did not confirm the clinical effectiveness, leading to withdrawal of atezolizumab by the FDA [139]. The reason for this is unclear but is the subject of much speculation [140]. Another checkpoint inhibitor study, KEYNOTE-355 [141], compared pembrolizumab (anti-PD1) with nab-paclitaxel, paclitaxel or carboplatin and gemcitabine in first-line therapy for metastatic TNBC patients and reported improved progression-free survival and OS in PD-L1-positive tumours (combined positive score (CPS) of >10) and led to the approval of pembrolizumab. Although approval for atezolizumab was withdrawn, a more detailed analysis of Impassion130 showed immune cell PD-L1 positivity was associated with clinical benefit, irrespective of TIL density. Furthermore, either germline or somatic *BRCA* mutation status was not associated with PD-L1 expression levels in immune cells [142], and the benefit of atezolizumab was noted in patients with PD-L1-positive immune cells, regardless of *BRCA* mutation status [142]; however, given the smaller numbers, it was suggested that further study is required in larger numbers to understand why this confirmatory trial was negative.

Innate immunity and Cyclic GMP–AMP synthase (cGAS)-stimulator of interferon genes (STING).

Increasingly recognised as a potent stimulator of anti-tumour immunity, activation of cyclic GMP–AMP synthase (cGAS)-stimulator of interferon genes (STING) results in a cascade sequence within the innate immune system, driving interferon production and heightening T-cell responses [143]. This pathway may be particularly significant in familial breast cancer, as compared to homologous recombination proficient tumours. In vitro studies of PARP inhibitors show a significant anti-tumour response dependent on *BRCA1*/2 deficiency, activation of the cGAS/STING pathway and recruitment of CD8+ T cells [144]. While the role players in the pathway are relatively well described, in several tumour streams, efficacy has been shown to be negatively impacted by predominantly downstream players, such as dendritic cell inactivity and PD-L1 activation, both of which were able to be overcome in vitro. While in vivo biomarkers have not yet been validated in this pathway, potential candidates include TANK-binding kinase 1 (TBK1), interferon regulatory factor 3 (IRF3) and IFN-related genes [145], the levels of the latter in the sera of a general cohort of 451 breast cancer patients being associated with poorer prognosis [146]. 

### 2.16. CXCL10/CXCR3

Several studies have shown a role for CXCL10, a key component of the Th1-associated immune response, as well as its receptor in the progression of several cancers, which are known to be significantly overexpressed in basal breast tumours [147,148]. Secreted by a host of immune (activated T lymphocytes, monocytes), stromal (fibroblasts, endothelial cells) and epithelial cells, the chemokine attracts other immune cells, including NK cells, with autocrine promotion of tumour growth also seen. In a study of familial breast cancers, tumour expression of both CXCL10 and the Th1-associated transcription factor T-BET was associated with higher tumour grade, higher proliferation index, tumour p53 expression, increased peritumour CD4+ and CD8+ lymphocytes. In relation to FOXP3-positive Tregs, there was no association, suggestive of independence from this immune-regulatory pathway. Furthermore, in the *BRCA1* subset, CXCL10-positive expression showed a worse prognosis, suggesting that the axis may serve as a potential target in these tumours [148]. 

## 3. Conclusions

While a lot of research has focused on cancer germline predisposition, there is a paucity of data relating to prognostic and predictive biomarkers in familial breast cancers. Study into this is inherently difficult in some instances, as *BRCA1*/2 germline mutation status is often not known before treatment begins. The identification of *BRCA*X patients is, in some instances, even more difficult, with few studies segregating these tumours from clear sporadic cancers. Recommended Guidelines, such as those produced by NICE, focus on proper identification of familial case, genetic screening and threshold for germline testing, surveillance of breast and other cancers and risk reduction. There are no treatment recommendations specific for familial breast cancers outside of generic breast cancer treatment guidelines. Similarly, the NCCN only provides guidelines for the assessment of high-risk/potential familial breast cancers that may require germline testing and testing panels. 

While cancers arising in *BRCA1*/2 mutation carriers certainly show marked sensitivity to platinum-based chemotherapeutic regimens and immunotherapies, there is still some heterogeneity in regards to treatment response. A small number of these tumours appear more phenotypically and genotypically similar to sporadic cancers, and they may require alternate therapies. Further evaluation is, therefore, required to identify possible cancers that may require alternate therapies. Furthermore, identification of treatment-resistant pathways is integral in either tailoring primary therapies or developing secondary and third-line treatment options.

Large studies into non-*BRCA1*/2 familial cancers are also needed. With ever-expanding familial databases and collaboration, further biomarkers in *PALB2* and *CHEK2* familial breast cancers should be explored, with some small case series showing some benefit of platinum-based chemotherapy in these cancers [149]. Description of *BRCA*X tumours has also been difficult with explorative studies general of modest size, with marked variability of inclusion criteria, and are often historically composed of genetic testing limited to *BRCA1*/2 only. These studies are universally retrospective and also show a lack of replication of findings, and frequently lack adequate controls. Certainly, there are some differences that are alluded to from sporadic cases of breast cancer. Whether there are specific predictive and prognostic biomarkers in this group, and whether they require alternate therapies, is, as yet, unknown. Prospective clinical trials evaluating novel combination therapies and identifying specific subgroups to most benefit from these treatments are warranted. Promising further work on immunotherapies in combination therapies with PIK3CA or PARP inhibitors [150] may also demonstrate additional benefits and potential new biomarkers. While almost all current studies asses these biomarkers at the time of cancer diagnosis, the optimal timing of the utility of these biomarkers is also somewhat unknown, with further studies required to assess promising targets in pre-clinical and in situ disease.

## Figures and Tables

**Figure 1 cancers-15-01346-f001:**
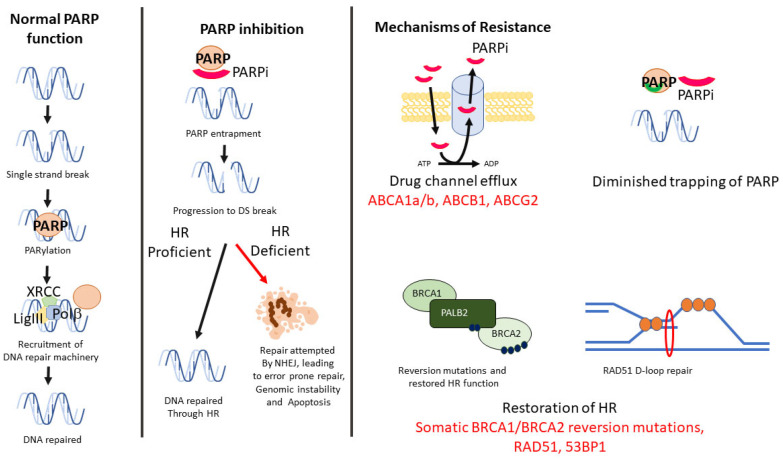
Mechanism of PARP, with repair of single-strand DNA break through PARPylation and recruitment of repair proteins of the X-ray repair cross-complementing protein 1 (XRCC1) complex. Inhibition of PARP leads to double-strand breaks, which may be repaired by HRR proteins if proficient, or apoptosis in HR-deficient cells. PARPi may occur through multiple mechanisms, including efflux of the inhibitor, mutation of PAPRP affecting trapping, reversion of HRR function or through RAD51-mediated repair, with potential biomarkers highlighted (in red).

**Figure 2 cancers-15-01346-f002:**
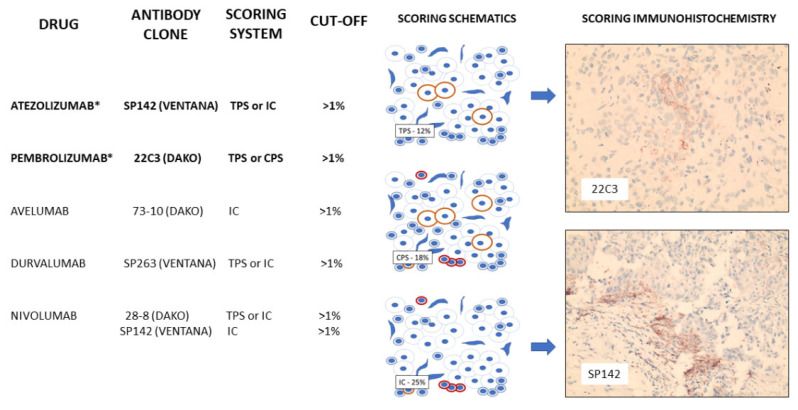
PD-L1 inhibitors currently studied in breast cancer, including those approved by the FDA for use in TNBC (*). Scoring schematics demonstrate counting methodology for tumour proportion score (TPS—percentage of tumour staining positively), immune score (IC—percentage of immune cells staining positively) and combine proportion score (CPS—percentage of tumour and immune cells staining positively) with patient immunohistochemical staining of tumour cells (22C3) and TILs (SP142) present.

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
