# Peer review of "Prognostic and Predictive Biomarkers in Familial Breast Cancer"

_cancers, 2023, doi:10.3390/cancers15041346_

Round 1

Reviewer 1 Report

What is the novelty of this study?

Please edit page 2, line 57 Homologous Recombination (HR)

PARP???

HER??

In figure 1. PARP? XRCC?.., Please explain these on the legend of figure 1.

MDR1????

Author Response

Thank you for the review of this article. We have replied to the points that were made.

What is the novelty of this study?

Currently the wider literature has largely focused on prognostic and predictive factors in all breast cancers in general, or biomarkers for genetic susceptibility, with no reviews present currently looking only at biomarkers only in familial breast cancers.

Please edit page 2, line 57 Homologous Recombination (HR)

This has been edited.

PARP???

PARP - poly ADP ribose polymerase

This has been expanded.

HER??

HER - human epidermal growth factor receptor

This has been expanded.

In figure 1. PARP? XRCC?.., Please explain these on the legend of figure 1.

The figure has been expanded and explained further.

XRCC - X-ray repair cross-complementing protein 1

MDR1????

MDR1 - Multi-Drug Resistance 1

This has been expanded.

Reviewer 2 Report

Review Report for the Manuscript “Prognostic and Predictive Biomarkers in Familial Breast Cancer

Rating the Manuscript

Quality of Presentation: Is the article written in an appropriate way? Are the highest standards for presentation of the results used?

Yes, the article is written well. Representation could be improved. Specially authors could use table and more figures like Figure 1, for better representation of the information in the manuscript.

Interest to the Readers: Are the conclusions interesting for the readership of the Journal? Will the paper attract a wide readership, or be of interest only to a limited number of people? (Please see the Aims and Scope of the journal)

Yes, this would be a great article for the researchers in the cancer research field.

English Level: Is the English language appropriate and understandable?

Yes, English language in the manuscript is appropriate and understandable. 

Overall Recommendation: Accept after Minor Revisions

Given below are the comments for each section of the manuscript.

Author list

Line 3: “Siddhartha Deb 1,2, Anannya Chakrabarti and Stephen B Fox4

The affiliation for the author Anannya Chakrbarthi is missing. Please include it.

Abstract

The abstract is written and summarizes the content of the manuscript.

Line 16: “Furthermore, tumours showing homologous recombination deficiency, due to loss of BRCA1, BRCA2, PALB2 and CHEK2 function have been shown to be especially sensitive to platinum-based chemotherapeutics and PARP inhibition”

It’s better if the authors could define the terms BRCA1, BRCA2, PALB2 and CHEK2 when they first appear in the manuscript.

2. PROGNOSTIC AND PREDICTIVE BIOMARKERS

Are all the prognostic and predictive biomarkers included in this section? If not, it would be a good idea to have a table summering all the biomarkers that have been discussed in the literature so far.

Figure 1: Nice figure, if possible, define the terms that have been used in the figure.

2.2. Non-BRCA associated Homologous Recombination

Line 134: “Similarly, a study was performed on HuR, an mRNA binding Hu family protein that enhance regulates translation of target transcripts through enhancing their stability (ref).”

Where’s the reference?

Line 136: “In many cancers, HuR levels are increased and oncogenic, primarily activated by mitogen-activated protein kinases (MAP-K).”

What are the other cancer types?

Line 141: “Notably, HuR levels were even higher in BRCA1 (63%) and BRCA2 (62%) tumours and associated with a non-significant reduction in survival in BRCA2 mutation carriers. There was no prognostic effect in the BRCA1 group. Based on these findings, HuR appears to be involved in the oncogenic pathways of familial breast cancer, probably through activation of the MAP kinase pathway, and is prognostic factor in at least some subsets of familial breast cancer.”

Authors need to add references to this section.

Line 150: “Although there was no difference in overall survival between patients with or without BRCA1/2 alterations, when stratifying by PALB2, PAGR1, RAD51B, FANCM, MLL4, or ERCC1/2a mutations, a small number of BRCA1- patients (n=27) with the additional a defect in HR genes had a significantly shorter survival”

It’s better if the authors could define the terms like PALB2, PAGR1, RAD51B, FANCM, MLL4, and ERCC1/2a when they first appear in the manuscript.

Line 158: 

2.3 HER

Define the term “HER”.

2.4. Hypoxia

Line 213: “HIF-1. overexpression correlates with poor prognosis in Triple Negative Breast Cancer (TNBC) including familial associated breast cancer (47, 48) (49, 50) (51, 52) (53)”.

Why are the references shown in pairs like this?

Line 215: “There are some data to suggest that BRCA1 modulates the hypoxic response by regulating HIF-1. stability and thereby vascular endothelial growth factor levels (see below)

(53).”

Instead of saying “see below”, it’s better if the authors mention section 2.5.

2.6. Cell Cycle Regulation

Line 239: “FOXM1 is a member of forkhead superfamily transcription factors and key regulator of cell cycle progression and DNA damage repair.”

Define the term “FOXMI”.

Line 240: “FOXM1 is over-expressed in most human cancers and predictive of poorer survival in breast cancer.”

In which cancer is FOXMI over-expressed?

2.7. Androgen Receptor

Line 248: “With ongoing development of AR targeted therapies showing clinical benefit (59), AR is a potential useful biomarker for these therapies.”

Briefly discuss about AR targeted therapies.

2.9. MicroRNA (miR)

“Although only a small number of specimens were used, Murria-Estal et al. (74). Reported 15 differentially expressed miRNAs that differentiates between BRCA1, BRCA2, BRCAX and sporadic breast tumours albeit with relatively low accuracy (75%).”

Briefly mention which miRNAs showed a significant change in expression.

Line 320: “Using prediction algorithms Moskwa et al. found miRNA-182 targeted BRCA1 in breast cancer (87) and overexpression of miRNA-182Its in MDA-MB231 cells were significantly more sensitive to PARP inhibitors (ref).”

Where’s the reference?

2.11. Commercial expression profile assays

It would be easy to follow if this information were included in a table with advantages and disadvantages of each commercial assay.  

2.14. Tumour infiltrating lymphocytes

Line 431: “Nevertheless, HRR deficiency is not always associated with high tumour mutational burden (ref).”

Where’s the reference?

Line 454: “Nevertheless, some studies have observed a direct correlation between increased numbers FOXP3 T regulatory cells and improved mortality which is somewhat unexpected given the immune regulatory role of these cells,and is contrary to the association with poor survival in breast cancer studies unselected for familial breast cancers (ref).”

Where’s the reference?

2.15. Programmed Cell Death ligand-1 (PDL-1)

Figure 2: Are these figures and information are taken from a reference? If so, the authors need to include the reference.

Author Response

Thank you for the review. The replies to the relevant questions are outlined below.

REVIEWER 2

Quality of Presentation: Is the article written in an appropriate way? Are the highest standards for presentation of the results used?

Yes, the article is written well. Representation could be improved. Specially authors could use table and more figures like Figure 1, for better representation of the information in the manuscript.

Very much representing the wider literature, the main areas of research in this field and wider clinical practice revolve around PARPi and immunotherapy, and we have therefore used these areas for our figures. Within the other areas, the data is very minimal or scant and we feel a comprehensive or useful table or figure may not be informative or useful.

Interest to the Readers: Are the conclusions interesting for the readership of the Journal? Will the paper attract a wide readership, or be of interest only to a limited number of people? (Please see the Aims and Scope of the journal)

Yes, this would be a great article for the researchers in the cancer research field.

English Level: Is the English language appropriate and understandable?

Yes, English language in the manuscript is appropriate and understandable. 

Overall Recommendation: Accept after Minor Revisions

Given below are the comments for each section of the manuscript.

Author list

Line 3: “Siddhartha Deb 1,2, Anannya Chakrabarti and Stephen B Fox4

The affiliation for the author Anannya Chakrbarthi is missing. Please include it.

This has been amended.

Abstract

The abstract is written and summarizes the content of the manuscript.

Line 16: “Furthermore, tumours showing homologous recombination deficiency, due to loss of BRCA1, BRCA2, PALB2 and CHEK2 function have been shown to be especially sensitive to platinum-based chemotherapeutics and PARP inhibition”

It’s better if the authors could define the terms BRCA1, BRCA2, PALB2 and CHEK2 when they first appear in the manuscript.

BRCA1 - BReast CAncer gene 1

BRCA2 - BReast CAncer gene 2

PALB2 - partner and localizer of the BRCA2 gene

CHEK2 - Checkpoint kinase 2

These have been defined further.

  1. PROGNOSTIC AND PREDICTIVE BIOMARKERS

Are all the prognostic and predictive biomarkers included in this section? If not, it would be a good idea to have a table summering all the biomarkers that have been discussed in the literature so far.

Figure 1: Nice figure, if possible, define the terms that have been used in the figure.

The figure has been expanded and explained further.

2.2. Non-BRCA associated Homologous Recombination

Line 134: “Similarly, a study was performed on HuR, an mRNA binding Hu family protein that enhance regulates translation of target transcripts through enhancing their stability (ref).”

Where’s the reference?

The reference has been added.

Line 136: “In many cancers, HuR levels are increased and oncogenic, primarily activated by mitogen-activated protein kinases (MAP-K).”

What are the other cancer types?

These have been added and include pancreatic exocrine adenocarcinoma, ovarian carcinoma, non-small cell lung carcinoma, colon carcinoma and upper genito-urinary carcinomas

Line 141: “Notably, HuR levels were even higher in BRCA1 (63%) and BRCA2 (62%) tumours and associated with a non-significant reduction in survival in BRCA2 mutation carriers. There was no prognostic effect in the BRCA1 group. Based on these findings, HuR appears to be involved in the oncogenic pathways of familial breast cancer, probably through activation of the MAP kinase pathway, and is prognostic factor in at least some subsets of familial breast cancer.”

HuR - Human antigen R

Authors need to add references to this section.

Line 150: “Although there was no difference in overall survival between patients with or without BRCA1/2 alterations, when stratifying by PALB2, PAGR1, RAD51B, FANCM, MLL4, or ERCC1/2a mutations, a small number of BRCA1- patients (n=27) with the additional a defect in HR genes had a significantly shorter survival”

The relevant reference was present earlier but has also been added to this sentence.

It’s better if the authors could define the terms like PALB2, PAGR1, RAD51B, FANCM, MLL4, and ERCC1/2a when they first appear in the manuscript.

PAGR1 - PAXIP1 Associated Glutamate Rich Protein 1

RAD51B - RAD51 Paralog B

FANCM - FA Complementation Group M

MLL4 - Myeloid/lymphoid or mixed-lineage leukemia 4

ERCC - Excision repair cross-complementing group

These have been defined further.

Line 158: 

2.3 HER

Define the term “HER”.

These have been defined further.

2.4. Hypoxia

Line 213: “HIF-1. overexpression correlates with poor prognosis in Triple Negative Breast Cancer (TNBC) including familial associated breast cancer (47, 48) (49, 50) (51, 52) (53)”.

Why are the references shown in pairs like this?

This has been amended.

 Line 215: “There are some data to suggest that BRCA1 modulates the hypoxic response by regulating HIF-1. stability and thereby vascular endothelial growth factor levels (see below)

(53).”

Instead of saying “see below”, it’s better if the authors mention section 2.5.

This has been amended and changed.

2.6. Cell Cycle Regulation

Line 239: “FOXM1 is a member of forkhead superfamily transcription factors and key regulator of cell cycle progression and DNA damage repair.”

Define the term “FOXMI”.

Factor Forkhead Box M1

Line 240: “FOXM1 is over-expressed in most human cancers and predictive of poorer survival in breast cancer.”

In which cancer is FOXMI over-expressed?

FOXM1 is overexpressed in multiple cancer type including epithelial cancers such as prostate adenocarcinomas, gastro intestinal cancers, non-small cell lung cancers, high -grade ovarian cancers,

2.7. Androgen Receptor

Line 248: “With ongoing development of AR targeted therapies showing clinical benefit (59), AR is a potential useful biomarker for these therapies.”

Briefly discuss about AR targeted therapies.

We have only very briefly included the relevant AR targeted therapies. We believe there are great articles recently published reviewing this area (Kolyvas et al 2022), and would be not as relevant for this paper.

2.9. MicroRNA (miR)

“Although only a small number of specimens were used, Murria-Estal et al. (74). Reported 15 differentially expressed miRNAs that differentiates between BRCA1, BRCA2, BRCAX and sporadic breast tumours albeit with relatively low accuracy (75%).”

Briefly mention which miRNAs showed a significant change in expression.

The list of 15 miRs have been added.

Line 320: “Using prediction algorithms Moskwa et al. found miRNA-182 targeted BRCA1 in breast cancer (87) and overexpression of miRNA-182Its in MDA-MB231 cells were significantly more sensitive to PARP inhibitors (ref).”

Where’s the reference?

The reference is Moskwa but has been properly formatted now.  

2.11. Commercial expression profile assays

It would be easy to follow if this information were included in a table with advantages and disadvantages of each commercial assay.  

The authors have also participated in presentation of this data in a more specific review and feel adding a table may conflict with that article with more considerable overlap than wanted.

2.14. Tumour infiltrating lymphocytes

Line 431: “Nevertheless, HRR deficiency is not always associated with high tumour mutational burden (ref).”

Where’s the reference?

This has been added and correctly formatted.

Line 454: “Nevertheless, some studies have observed a direct correlation between increased numbers FOXP3 T regulatory cells and improved mortality which is somewhat unexpected given the immune regulatory role of these cells,and is contrary to the association with poor survival in breast cancer studies unselected for familial breast cancers (ref).”

Where’s the reference?

This has been added and correctly formatted.

2.15. Programmed Cell Death ligand-1 (PDL-1)

Figure 2: Are these figures and information are taken from a reference? If so, the authors need to include the reference.

This figure is original to the article.

Reviewer 3 Report

The Authors undertook important topic in current oncology. They tried to assess prognostic and predictive biomarkers in familial breast cancer. The article is an extensive elaboration. Many biomarkers are described but with limited utility. The Authors often evoke research on cell lines with limited use.

Please consider to add more practical data. In some sections there are only descriptions of mechanisms of potential interactions.   

Overall, this manuscript is generally well-written and the topic is important and appropriate for Cancers and its audience. The manuscript could be published after changes.

Author Response

THANK YOU FOR THE REVIEW AND PERTINENT COMMENTS.

CURRENTLY, THERE IS VERY LIMITED KNOWLEDGE OF BIOMARKES IN FAMILIAL BREAST DUE TO:  ABSENCE OF PROSPECTIVE STUDIES, AND A LACK OF CLASSIFICATION OF TUMOURS INTO SPORADIC AND FAMILIAL GROUPS OFTEN DUE TO DELAY IN ESTABLISHING GERMLINE MUTATION OR SIGNIFICANT FAMILIAL HISTORY. HENCE THERE IS A MARKED ABSENCE OF BOTH PRACTICAL DATA, AND CLINICAL STUDY. WE HAVE INCLUDED ALL SIGNIFICANT RELEVANT STUDIES PERFORMED TO DATE IN FAMILIAL BREAST CANCER BUT HAVE ALSO ATTEMPTED TO BRING TO SIGNIFCANCE POSSIBLE PRE-CLINICAL STUDIES THAT DEMONSTRATE POTENTIAL AREAS WHERE BIOMARKERS MAY BE IMPORTANT. THE LACK OF FURTHER PRACTICAL DATA IS NOT INTENTIONAL BUT DUE TO AN ABSENCE OF CORRELATIVE CLINICAL STUDIES. WHERE POSSIBLE WE HAVE ATTEMPTED TO SHOW THE RELEVANT RELATIONSHIP IN OTHER CANCER STREAMS.

Reviewer 4 Report

This is a good review, focused on prognostic and predictive biomarkers for familial breast cancer.

Please add a suggestion for when is the best time for women or men to start checking those biomarkers.

Author Response

Thank you for your review and relevant question. As this study is looking at tumour biomarkers in familial breast cancer, the time to check these would be following the diagnosis of cancer. While this is not a review of biomarkers of tumour susceptibility in familial breast cancer, we agree that the question of biomarker testing may be important particularly in pre-clinical and in situ disease. We have subsequently added a line in the conclusion pertaining to this.

Round 2

Reviewer 3 Report

Dear Authors,

Thank you for your reply. I appreciate your extensive work. I like changes you did in "simple Summary".

Thank you!